# A Novel Facet of In-Hospital Food Consumption Associated with Hospital Mortality in Patients with Scheduled Admission—Addition of a Study Protocol to Test the Existence of Effects of COVID-19 in the Same Study in the Post-COVID-19 Period

**DOI:** 10.3390/nu16142327

**Published:** 2024-07-19

**Authors:** Hiroyo Miyata, Ayako Tsunou, Yoko Hokotachi, Teruyoshi Amagai

**Affiliations:** 1Administration Food Sciences and Nutrition Major (Doctoral Program), Graduate School of Human Environmental Sciences, Mukogawa Women’s University, Nishinomiya 663-8558, Japan; yu111164@yahoo.co.jp (H.M.); ayako.tsunou@gmail.com (A.T.); stgrw766@ybb.ne.jp (Y.H.); 2Department of Clinical Nutrition, Kindai University Hospital, Osaka 589-8511, Japan; 3Department of Clinical Nutrition, Kitauwa Hospital, Uwajima 798-1392, Japan; 4Department of Clinical Nutrition, Takarazuka Dai-Ichi Hospital, Takarazuka 665-0832, Japan; 5Department of Clinical Engineering, Faculty of Health Care Sciences, Jikei University of Health Care Sciences, Osaka 532-0003, Japan

**Keywords:** single-day survey, hospital mortality, hospital food consumption, scheduled admission, seasonality

## Abstract

Background: Humankind has faced unexperienced pandemic events since 2020. Since the COVID-19 pandemic has calmed down, we felt the need to verify whether in-hospital mortality had worsened compared to pre-pandemic conditions due to the COVID-19 pandemic. Objective: To test the hypothesis that daily food consumption is associated with in-hospital mortality during hospitalization and to provide baseline data to examine whether the effects of COVID-19 exist or not in post-pandemic period. Methods: All hospitalized patients staying in a single institution on the third Thursday of May, August, November, and the following February were included. Compared data: (1) among four seasons, (2) between age < 75 vs. ≥75 years, (3) between <75% vs. ≥75% of in-hospital food, and (4) logistic regression analysis to identify factors associated with in-hospital mortality. Results: In 365 inpatients, the following results were obtained: (1) no seasonality or age effect in in-hospital mortality, (2) the novel cutoff value of 75% of the hospital food requirement was used to identify poor in-hospital survivors, (3) logistic regression analysis showed low food consumption, with <75% of the hospital food requirement as the predictor of high in-hospital mortality. Conclusions: A small eater of in-hospital food < 75% during hospitalization was associated with significantly higher in-hospital mortality in patients with scheduled hospitalization in the pre-pandemic period. Then, a study protocol is proposed to test the existence of the effects of COVID-19 in the same study in the post-COVID-19 period. This study protocol is, to our knowledge, the first proposal to test the effects of food consumption in the post-COVID-19 period on in-hospital mortality in the clinical nutritional areas.

## 1. Introduction

The total number of worldwide COVID-19 cases reported to the WHO is 686,632,076, and the total number of deaths is now 6,855,479, as of June 2024 [1]. This statistic is based on a world population of 811,900,000, so it can be said that 8.5% of the world population has been infected with COVID-19, 1% of infected patients have died, and the remaining 99% have survived. While 99% of patients survived and returned to society after COVID-19 infection, it is necessary to verify whether a history of COVID-19 infection affects post-pandemic outcomes. Therefore, in this study, we would like to provide baseline data obtained in the pre-pandemic period for the post-pandemic period on factors influencing in-hospital mortality of scheduled admission patients to determine whether the COVID-19 pandemic has impacted the in-hospital mortality of these inpatients. To accomplish this purpose, we analyzed predictors of outcomes for inpatients with scheduled admissions obtained in the pre-COVID period in a one-day-per-season cross-sectional survey across four seasons.

### 1.1. Findings of the Previous Study, Which Showed an Association between In-Hospital Food Consumption and In-Hospital Mortality in Patients with Emergency Admission

Before completing this study of planned hospital admissions, we paid attention to food consumption in hospitals because of several findings: (1) A 30-day follow-up study of hospitalized patients in 31 units of a Danish university hospital showed that an increased risk of mortality was associated with patients who did not consume 75% of the estimated requirement (energy: OR ¼ 8.08 [1.78; 36.79]; protein: OR ¼ 3.40 [0.74; 15.53]) [2]. (2) Similar findings have been reported in patients with chronic obstructive pulmonary disease (COPD) [3]. (3) This may be due to increased inflammation, which is closely associated with poor appetite and reduced food intake [4].

Consistent with our expectation that hospital food consumption is associated with outcomes, we conducted the same styled study as the present study to determine the same effects on in-hospital mortality in patients with emergency admission for four seasons. In this study, significantly higher in-hospital mortality was found in the following patients: (1) emergency versus scheduled admissions, (2) in all four seasons, and (3) whose in-hospital food consumption was <75% of the requirement [5,6].

However, contrary to our prediction, no seasonal differences in in-hospital mortality were observed when comparing emergency admission data across four seasons. In-hospital mortality in emergency patients is independent of season.

The present study aims to determine whether the finding that smaller eaters with higher mortality is limited to emergency patients or also occurs in elective patients. If so, it will be found that reduced dietary consumption is an important predictor of in-hospital mortality regardless of the type of hospitalization. Conversely, if this finding was not observed in scheduled admitted patients, why is in-hospital food consumption not a predictor of in-hospital mortality only in emergent admitted patients? This study was conducted to address these clinical questions.

### 1.2. Objective

The purpose of this study is to test the hypothesis that daily food consumption during hospitalization of patients scheduled for hospitalization is associated with in-hospital mortality in a one-day cross-sectional survey across four seasons.

## 2. Materials and Methods

### 2.1. Participants

All inpatients staying in hospital on the study day were eligible for the study. Data were collected from the subjects’ electronic medical records. The details of the collected data and their timing of collection during hospitalization are shown in Appendix A.

Food consumption measurement was conducted by the ward nurses who collected the completed meals; they assessed the amount of food consumption by visually estimating 100%, 75%, 50%, and 25%. In-hospital mortality was defined as death at the time of discharge.

The subjects were divided into four subgroups according to the season of the study: S1, S2, S3, and S4, for groups of spring, summer, autumn, and winter, respectively (Appendix A).

### 2.2. Study Design

An international cross-sectional point survey of patients with or at risk of malnutrition [7,8,9], called the nutritionDay (nDay), has been conducted once a year. This nDay is held worldwide on a fixed date, the third Thursday of November. However, our concern was that the results of a survey conducted on a single day in November might be different depending on the season. To address this concern, we conducted additional one-day cross-sectional surveys in the other three seasons, in addition to the November survey, using the same conditions as for nDay. The study dates for the three seasons other than November in this study, similar to the third Thursday as the original day set in nDay, were set to the third Thursday of May, August, and February of the following year for the three seasons. The reason for choosing the Thursday was to avoid weekend effects.

All data collected were analyzed using the following four methods. The details of these methods were the same as our previously published study [5]. Exclusion criteria included patients in the following three categories, which were the same as in the previously published analysis: (1) age < 18 years old, (2) admission style was emergency, non-scheduled, (3) any patients with missing data.

### 2.3. Four Seasons’ Analysis

Data were compared separately among the four seasons to test for seasonal differences in subject demographics, in-hospital food consumption, and outcome measures such as in-hospital mortality.

In subsequent analyses, all four groups of S1, S2, S3, and S4 (ΣS in Appendix A) were pooled to proceed to Section 2.4 and Section 2.5. In the following Section 2.4 and Section 2.5, to confirm whether there were differences in outcomes, such as in-hospital motility according to age and food consumption on the study day, all subjects were divided into two groups according to age (Section 2.4) and in-hospital food consumption (Section 2.5).

### 2.4. Age Analysis

All subjects were divided into two subgroups by age, 75 years or older and under 75 years, then all data were compared.

### 2.5. Food Consumption Analysis

All subjects in group ΣS, the combined group of all four seasons, were divided into two subgroups according to in-hospital food consumption with respect to two cutoffs set at 50% and 75% of the requirement in methods 3-1 and 3-2, respectively. Here, we set the cutoffs at 50% and 75% because they seemed relatively easy to assess at a glance, such as half or three-quarters of a whole serving of in-hospital food, and were considered clinically feasible.

### 2.6. Multivariate Analysis

A multivariate logistic regression analysis was performed to identify the predictive factors for the in-hospital mortality of patients in group ΣS.

### 2.7. Statistical Analysis

Data are presented as median, 25th percentile, and 75th percentile. The Mann–Whitney U test was used for differences in median values between two groups, the Kruskal–Wallis test was used for comparisons between four groups, and the χ^2^ test or Fisher’s exact test was used for differences in proportions between groups. Multivariate logistic regression analysis with adjusted odds ratios for in-hospital mortality was performed. The adjusted odds ratios, 95% confidence intervals, and their *p*-values represent the odds of in-hospital mortality after adjusting for the covariates listed in the table. A statistically significant difference was considered significant when *p* < 0.05. Statistical analysis was performed with SPSS version 29 (IBM, Armonk, NY, USA).

## 3. Results

The total number of hospitalized patients on the four study days of the four seasons was 671. There were 38 cases under the age of 18, 50 cases with missing data, and 218 cases with emergency admission. After excluding all these cases, 365 cases remained. These cases proceeded in the further analyses in method 1 and thereafter (Figure 1).

### 3.1. Results of Section 2.3

In comparing all collected data among the four groups in four seasons, not only demographics and in-hospital food consumption amounts, but in-hospital mortality and the length of stay in hospital were not significantly different. In addition, no significant differences were observed in all aspects, including CRP as the index of inflammation (Table 1). This finding is interpreted to mean that there is no seasonality in in-hospital mortality in patients with scheduled admission.

### 3.2. Results of Section 2.4

Assuming that an outcome such as in-hospital mortality is associated with older age, we examined the effect of age by comparing the two subgroups between those aged 75 years and older and those younger than 75 years. As a result, no significant differences were observed in any of the outcome measures, such as in-hospital mortality (Table 2).

### 3.3. Results of Section 2.5

To clarify the extent to which in-hospital nutritional inadequacy affects in-hospital mortality, we established two cutoff values for nutritional inadequacy: 50% and 75% of hospital food, in Section 2.5. In both analyses, the CCI, ADL, in-hospital mortality, and CRP (the highest during hospitalization) were all significantly worse in subjects whose in-hospital food consumption was less than 50% (Table 3) and 75% (Table 4), respectively.

In these two analyses, considering a priori that 75% ingestion is detected earlier than 50% consumption, we set the cutoff value at 75% to detect patients with poor outcomes earlier.

### 3.4. Results of Section 2.6

In-hospital mortality and factors associated with in-hospital food consumption < 75% were analyzed using logistic regression analyses. The factors associated with in-hospital mortality were the CCI and in-hospital food consumption (Table 5).

In addition, the factors associated with food consumption < 75% identified in method 3 were female and had a higher CCI of comorbidity (Table 6).

## 4. Discussion

### 4.1. The Need to Examine Whether the COVID-19 Pandemic Had an Effect on In-Hospital Mortality Due to COVID History

Based on worldwide statistics, 8.5% of the world population has been infected with COVID-19, 1% of infected patients have died, and the remaining 99% have survived [1]. In other words, it can be assumed that 99% of the patients saved will continue to be hospitalized with a history of COVID-19.

The reason for conducting this study was the need to verify whether in-hospital mortality had worsened, compared to pre-pandemic conditions, due to the COVID-19 pandemic, which has been an unprecedented human event. If the result in the post-pandemic period is different from that in the pre-pandemic period, it would indicate that the pandemic had an effect on mortality, and its association factor might include food consumption; therefore, the cutoff value for food consumption needs to be reassessed.

### 4.2. Which Nutritional Assessment Tool Was Most Used during the COVID-19 Pandemic and Which Tools Include Food Consumption Assessment?

A total of 1743 articles were collected studying the nutritional assessment tools in patients with COVID-19 searched on June 2024 through the PubMed platform. In these articles, assessment tool number NRS2002 was used the most, followed by MUST, MNA-SF, GLIM, PNI, and GNRI (Figure 2 and Appendix A, with 44 references added in Supplementary references, excluding those listed at the end of the article).

Looking at Figure 2 (and Appendix A), the number of publications using nutritional assessment tools for COVID-19 patients was the highest for the NRS2002, with 17. The NRS2002 is the only assessment tool with a cutoff value of 75% of the in-hospital food consumption obtained in this study and may be the optimal tool for early detection of malnutrition in COVID-19 patients. The NRS2002 consists of two steps to assess nutritional status, and food consumption is assessed in the second stage after the first-stage assessment is completed. Given that the results of our study showed that 75% of dietary consumption is relevant in predicting mortality, food consumption assessment with a cutoff value at 75% must be moved to the first stage in the NRS2002. In our previous study, the same result was proved [5]. Another analysis also stated that low in-hospital food consumption is associated with in-hospital mortality [8]. However, these were limited to emergency admission or subspecialty wards, such as surgical [8,10,11]. To the best of our knowledge, this study is the first to successfully prove that a smaller eater, of less than 75% of hospital food, is associated with high in-hospital mortality, regardless of admission style.

### 4.3. A Novel Cutoff Value of In-Hospital Food Consumption Setting at 75% of Requirements

The analysis in Figure 2 shows that the nutritional screening tools used in the pandemic period were all the same as those used in the pre-pandemic period. Among these assessment tools, all of them did not apply the cut-off value set at 75%, except for the NRS 2002. From our results that the cutoff value of food consumption set at 75% seems to be predictive of hospital mortality, the cutoff value of in-hospital food consumption for inpatients must be set at 75% to detect malnutrition earlier. In various nutritional screening tools, the cutoff value of food consumption associated with malnutrition is set at varying from 50% to 75% of the requirement in GLIM and NRS2002 [12,13], respectively. The other tools do not describe a specific food consumption decrease, such as subjective global assessment (SGA) [14], MNA [15], MNA-SF [16,17], and MST [18]. As noted in our research [5], in-hospital food consumption is an important predictor of in-hospital mortality. Setting the cutoff for in-hospital food consumption at 75% instead of 50% theoretically allows for earlier identification of poor outcomes and earlier initiation of nutritional intervention. This is consistent with the results of the previous study in the emergency admitted patients. Therefore, it could be concluded that a smaller eater of <75% of in-hospital food is a predictor of in-hospital mortality, regardless of the admission style or season.

Based on the results of this study, we propose that in the future, all hospitalized patients who consume less than 75% of the hospital food provided should be considered nutritionally at risk, i.e., patients at a high risk for high in-hospital mortality. We propose to develop a nutritional policy that examines (1) the reasons for low food consumption, and (2) if food consumption above 75% is not possible, worsening inflammatory symptoms must be recognized and treated earlier to prevent in-hospital mortality.

Looking at the previously published studies, when the cutoff value was set at 75% of the energy and protein requirements, ORs of longer stays in hospital were 1.59 (95% confidence interval, 1.49–1.70) and 1.36 (95% CI, 1.27–1.45), respectively [19]. However, this analysis did not achieve the deference of mortality, which must be stronger than the length of hospitalization. Another study also analyzed that small eaters of the offered in-hospital food < 75% was associated with two times higher CRP [20].

The outcome measures in these reports were length of hospital stay and CRP. However, there was one other study that examined the relationship between mortality and 75% food intake [3]. The subjects were patients with chronic obstructive pulmonary disease (COPD), with a median age of 75 years, but no association between mortality and disease was found. The difference between our study and this one is that the subjects were limited to COPD patients who had a long history of chronic malnutrition associated with COPD, and the median mean induction period was short, at 9.0 days (95% CI, 9.0–11.0). In comparison to these studies, our study included older subjects with many comorbidities, and in subjects who were already severely malnourished, food intake < 75% of the requirement is considered a predictor of mortality.

### 4.4. Strength and Limitations of the Present Study

The strength of this paper is that it showed that total food consumption of three meals per day was a factor associated with in-hospital mortality. In addition, the 75% cutoff was shown to have the potential to detect malnutrition earlier than the traditional 50% a priori.

It is also warranted to highlight the weaknesses of this study. First, we have not taken into account the other cutoff values, such as 80% or 90%, that would allow us to detect malnutrition even earlier. However, it is currently unclear whether it is possible to clearly distinguish between 75% and 80% or 90% and whether this 5% or 15% difference affects the outcome. Second, there is no guarantee that the newly proposed cutoff of 75% of the requirement for in-hospital food consumption will detect malnutrition and poor outcomes as early as the 50% cutoff. Further study is needed to verify this. Third, it is unclear whether the same or worse results are obtained on weekends, as this survey was conducted on a weekday to avoid weekend effects. It is also unclear whether the results obtained this time apply to all weekdays or are limited to Thursdays. Fourth, as with the day of the week issue, there is also a seasonal issue. As to whether the same results would be obtained in months other than January of each season in which the one-day survey was conducted this time (February, May, August, and November), we did not examine the eight other months, so the results are limited to January of each season in which the survey was conducted. The additional eight untested months need to be surveyed. Lastly, the results are too small to draw conclusive findings. Further large prospective studies on other weekdays or weekends are needed.

### 4.5. Proposal of Study Protocol

The results of our current and previous studies show that before the pandemic, in-hospital mortality was poor for hospitalized patients admitted on weekdays, regardless of whether the admission was scheduled or emergency, and regardless of the four seasons.

Malnutrition due to sarcopenia has also been reported in the post-COVID period [21,22]. However, it is necessary to verify in the post-COVID period whether this is actually due to the direct effects of COVID, or whether food consumption has actually decreased to less than 75%.

To clarify this, we propose a protocol to test whether this result has changed after the COVID-19 pandemic, similar to this study. If this verification shows that the OR of in-hospital food consumption on in-hospital mortality is high, it suggests the possibility that the COVID-19 pandemic may have had long-term effects on in-hospital mortality.

As a post-COVID effect, a decline in cardiopulmonary function due to pulmonary hypertension and right ventricular dysfunction [23,24,25] has been reported. This may well lead to a decrease in food intake during hospitalization. However, no studies have been conducted on food consumption < 75% as an outcome or on in-hospital mortality.

The proposed study protocol is exactly the same as this study, including scheduled and emergency admission. Furthermore, the data collected and the outcome measures are the same. Then, the ORs of food consumption on in-hospital mortality between pre- and post-COVID-19 mortality are compared.

As for the results of the proposed study protocol, if the OR of the post-pandemic is higher than that of the pre-pandemic, as shown in this study, this must show the existence of the effects of food consumption on mortality and vice versa. This study protocol is, to our knowledge, the first to test the effects of food consumption in the post-COVID-19 period on in-hospital mortality in the clinical nutritional areas.

## 5. Conclusions

A small eater of in-hospital food < 75% during hospitalization was associated with significantly higher in-hospital mortality in patients with scheduled hospitalization in the pre-pandemic period. Then, a study protocol was proposed to test the existence of the effects of COVID-19 in the same study in the post-COVID-19 period.

## Figures and Tables

**Figure 1 nutrients-16-02327-f001:**
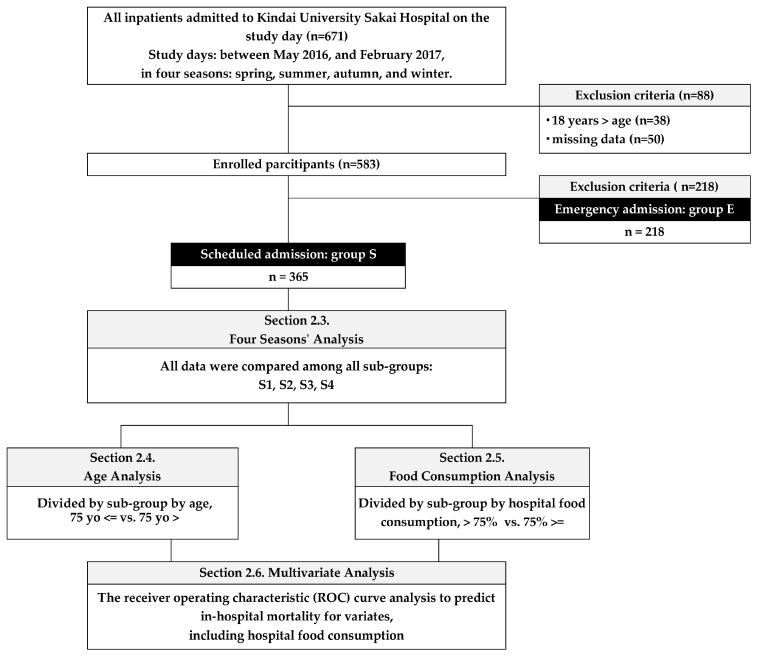
The flow diagram of the study. Abbreviations, E: Patients with emergency admission, S: Patients with scheduled admission.

**Figure 2 nutrients-16-02327-f002:**
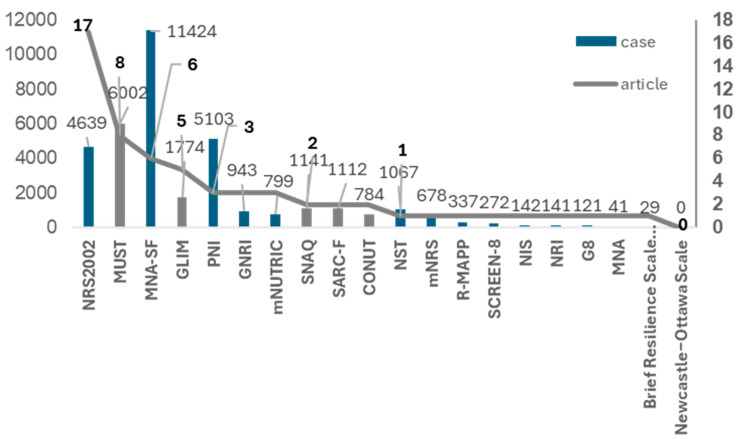
The list of nutritional assessment tools for COVID-19 patients. NRS 2002 had the largest number of papers, while MNA-SF had the largest number of patients. Among them, NRS2002 was the only one that set the cutoff for food consumption at 75%. It was also suggested that NRS2002 would be suitable to test whether food consumption has an effect on COVID-19 mortality in the post-COVID-19 period.

**Table 1 nutrients-16-02327-t001:** The results of four seasons analysis in Section 2.3.

Subgroups of Four Seasons	Group S1	Group S2	Group S3	Group S4	*p*-Value
Demographics					
Sex, male, N (%)	50 (56)	45 (49)	45 (46)	47 (55)	0.468
Age, years	73 (65, 79)	72 (62, 79)	71 (65, 77)	72 (64, 76)	0.797
BMI, kg/m^2^	22.9 (19.5, 25.5)	22.3 (19.8, 24.5)	22.2 (20.3, 25.6)	22.1 (19.8, 24.4)	0.892
Admitted ward, N (%) of internal medicine	50 (56)	54 (59)	62 (63)	56 (66)	0.497
CCI score	4 (2, 6)	4 (2, 6)	4 (1, 5)	4 (2, 6)	0.558
Walking as ADL, N (%)	72 (80)	75 (82)	82 (84)	73 (86)	0.750
Number of drugs, types	6 (3, 10)	5 (2, 8)	5 (2, 9)	5 (3, 8)	0.614
LOS before study day	9 (2, 15)	3 (1, 16)	8 (2, 17)	10 (2, 17)	0.276
Nutritional parameters					
Hospital food intake, %	95 (67, 100)	84 (57, 100)	83 (65, 100)	85 (64, 100)	0.591
% food intake, N (%) of patients taking hospital food ≥ 75%	59 (66)	54 (59)	64 (65)	49 (58)	0.563
Food texture by IDDIS, N (%) of participants taking regular diets *	69 (90)	57 (79)	76 (93)	68 (96)	0.008
Outcome parameters					
Primary outcome					
In-hospital mortality, N (%)	5 (3)	2 (2)	1 (1)	2 (2)	0.271
Secondary outcomes					
LOS, days	20 (12, 36)	17 (6, 43)	19 (10, 33)	22 (11, 38)	0.577
Highest CRP around study day, mg/dL	2.2 (0.3, 6.1)	1.1 (0.2, 6.1)	1.4 (0.3, 5.7)	2.2 (0.3, 7.7)	0.789
Survival within 30 days after hospitalization, N (%)	90 (99)	92 (100)	98 (100)	85 (100)	0.383

Comparing all collected data among four groups in four seasons, not only demographics and amounts of in-hospital food consumption, but in-hospital mortality as the primary outcome measure and the length of stay in hospital as the second one, were not significantly different among four season groups. From these findings, in the case of patients with scheduled admission, the subjects of this study, there are not only no seasonal differences in age, body size, and severity of comorbidity (CCI), but also no seasonal differences in in-hospital mortality outcomes. Analyses with * exclude subjects who missed lunch. Abbreviations, ADL: Activity of Daily Living, BMI: Body Mass Index, CCI: Charlson Comorbidity Index, CRP: C-reactive protein, IDDSI: International Dysphagia Diet Standardisation Initiative, LOS: Length of stay in hospital.

**Table 2 nutrients-16-02327-t002:** Results of age analysis in Section 2.4.

	Age < 75	Age ≥ 75	*p* Value
Number of subjects	236	129	
Demographics			
Sex, male, N (%)	126 (53)	61 (47)	0.265
Age, years	67 (59, 72)	79 (77, 82)	<0.001
BMI, kg/m^2^	22.6 (20.1, 25.3)	21.6 (19.5, 24.9)	0.062
Admitted ward, N (%) of internal medicine	145 (61)	77 (60)	0.743
CCI score	4 (2, 6)	4 (3, 6)	0.248
Walking as ADL, N (%)	204 (86)	98 (76)	0.011
Number of drugs, types	5 (2, 8)	8 (4, 11)	<0.001
LOS before study day	8 (2, 15)	8 (2, 21)	0.835
Nutritional parameters			
Food texture by IDDIS, N (%) of participants taking regular diets *	173 (89)	97 (90)	0.863
Hospital food consumption, %	92 (65, 100)	75 (54, 97)	0.131
Outcome parameters			
Primary outcome			
In-hospital mortality, N (%)	5 (2)	5 (4)	0.254
Secondary outcomes			
LOS, days	22 (12, 38)	26 (16, 45)	0.249
Highest CRP during the entire hospital study, mg/dL	1.8 (0.2, 6.3)	1.4 (0.4, 5.5)	0.850
Survival within 30 days after hospitalization, N (%)	236 (100)	128 (99)	0.353

After combining all data of four seasons because there were no seasonal differences among them as shown in result 2, all description parameters of patients with subgroup of aged ≥75 years and <75 years old were not significantly different. Not only that, in-hospital mortality as the primary outcome, the length of stay (LOS), the highest CRP during the entire hospital stay, and the death rate during hospitalization within 30 days were also not significantly different. From these results, in patients with scheduled admission, we found that there were no differences in any characteristics other than age, food intake, or outcome measures such as in-hospital mortality between the two subgroups: those aged 75 years or older and those younger than 75 years. In other words, in patients with a scheduled admission, age is not associated with outcome in this study, contrary to what was previously thought. Analyses with * exclude subjects who missed lunch. Abbreviations, ADL: Activity of Daily Living, BMI: Body Mass Index, CCI: Charlson Comorbidity Index, CRP: C-reactive protein, IDDSI: International Dysphagia Diet Standardisation Initiative, LOS: Length of stay in hospital.

**Table 3 nutrients-16-02327-t003:** Results of hospital food consumption with cutoff value set at 50% in Section 2.5. In setting the cutoff of hospital food intake at 50% in Section 2.5, CCI, ADL of walking ability, in-hospital mortality, and CRP (the highest during hospitalization) were all worse in patients with less than each cutoff. Analyses with * exclude subjects who missed lunch. Abbreviations, ADL: Activity of Daily Living, BMI: Body Mass Index, CCI: Charlson Comorbidity Index, CRP: C-reactive protein, IDDSI: International Dysphagia Diet Standardisation Initiative, LOS: Length of stay in hospital.

	Hospital Food Intake		
	<50%	≥50%	*p* Value
Number of subjects	308	57	
Demographics			
Sex, male, N (%)	157 (51)	30 (53)	0.818
Age, years	72 (64,77)	74 (64, 79)	0.971
BMI, kg/m^2^	22.6 (19.9, 25.3)	21.3 (18.2, 23.3)	0.039
Admitted ward, N (%) of internal medicine	183 (59)	39 (68)	0.201
CCI score	4 (2, 6)	5 (3, 8)	<0.001
Walking as ADL, N (%)	267 (87)	35 (61)	<0.001
Number of drugs, types	6 (3, 9)	7 (3, 11)	0.719
LOS before study day	8 (2, 15)	6 (1, 24)	0.964
Nutritional parameters			
Food texture by IDDIS, N (%) of participants taking regular diets *	257 (91)	13 (72)	0.031
Hospital food consumption, %	95 (72, 100)	5 (0, 29)	<0.001
Outcome parameters			
Primary outcome			
In-hospital mortality, N (%)	2 (1)	8 (14)	<0.001
Secondary outcomes			
LOS, days	24 (15, 41)	23 (8, 46)	0.765
Highest CRP during the entire hospital study, mg/dL	1.4 (0.2, 5.3)	3.8 (0.7, 11.3)	0.015
Survival within 30 days after hospitalization, N (%)	308 (100)	56 (98)	0.156

**Table 4 nutrients-16-02327-t004:** Results of hospital food consumption with cutoff value set at 75% in Section 2.5. In setting the cutoff of hospital food intake at 75%, the same results were shown as in Table 3. Comparing these two results of Section 2.5, setting the cutoff at 75% is expected to provide earlier identification of malnutrition, leading to early nutritional intervention and improved outcome. Analyses with * exclude subjects who missed lunch. Abbreviations, ADL: Activity of Daily Living, BMI: Body Mass Index, CCI: Charlson Comorbidity Index, CRP: C-reactive protein, IDDSI: International Dysphagia Diet Standardisation Initiative, LOS: Length of stay in hospital.

	Hospital Food Intake		
	<75%	≥75%	*p*-Value
Number of subjects	226	139	
Demographics			
Sex, male, N (%)	118 (52)	69 (50)	0.633
Age, years	71 (64,76)	74 (66, 79)	0.948
BMI, kg/m^2^	22.6 (20.2, 25.4)	21.8 (19.0, 24.6)	0.077
Admitted ward, N (%) of internal medicine	129 (57)	93 (67)	0.062
CCI score	4 (2, 5)	5 (3, 7)	0.003
Walking as ADL, N (%)	198 (88)	104 (75)	0.002
Number of drugs, types	6 (3, 9)	6 (4, 10)	0.266
LOS before study day	9 (3, 16)	3 (1, 19)	0.002
Nutritional parameters			
Food texture by IDDIS, N (%) of participants taking regular diets *	207 (92)	63 (83)	0.033
Hospital food consumption, %	100 (90, 100)	56 (12, 67)	<0.001
Outcome parameters			
Primary outcome			
In-hospital mortality, N (%)	0 (0)	10 (7)	<0.001
Secondary outcomes			
LOS, days	24 (15, 38)	24 (11, 45)	0.974
Highest CRP during the entire hospital study, mg/dL	1.2 (0.2, 4.6)	2.8 (0.5, 11.3)	<0.001
Survival within 30 days after hospitalization, N (%)	226 (100)	138 (99)	0.381

**Table 5 nutrients-16-02327-t005:** Results of logistic regression analysis to find factors associated with in-hospital mortality. Among the seven variants analyzed, CCI and in-hospital food consumption were significant and OR had a significant *p*-value. Abbreviations, ADL: Activity of Daily Living, BMI: Body Mass Index, CCI: Charlson Comorbidity Index, CI: confidence interval, IDDSI: International Dysphagia Diet Standardisation Initiative, OR: odds ratio.

Variables	Reference	OR (95% CI)	*p* Value
Sex	male	0.362 (0.047–2.762)	0.327
Age		0.997 (0.923–1.077)	0.946
BMI		1.130 (0.841–1.519)	0.417
CCI		1.485 (1.021–2.160)	0.039
ADL	walking	1.059 (0.099–11.312)	0.962
Food texture by IDDIS	regular diets	2.417 (0.245–23.835)	0.450
Hospital food consumption		0.450 (0.288–0.703)	<0.001

**Table 6 nutrients-16-02327-t006:** Results of logistic regression analysis to find factors associated with in-hospital food consumption less than 75% of requirement. Among the six variants analyzed, CCI was significant and OR had a significant *p*-value. Abbreviations, ADL: Activity of Daily Living, BMI: Body Mass Index, CCI: Charlson Comorbidity Index, CI: confidence interval, IDDSI: International Dysphagia Diet Standardisation Initiative, OR: odds ratio.

Variables	Reference	OR (95% CI)	*p* Value
Sex	male	1.838 (1.072–3.155)	0.027
Age		0.999 (0.979–1.019)	0.931
BMI		0.948 (0.887–1.012)	0.110
CCI		1.134 (1.030–1.248)	0.011
ADL	walking	1.466 (0.713–3.012)	0.298
Food texture by IDDIS	regular diets	1.595 (0.698–3.636)	0.268

## Data Availability

The data presented in this study are available on request from the corresponding author due to ethical reasons.

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
