# Peer review of "A Novel Facet of In-Hospital Food Consumption Associated with Hospital Mortality in Patients with Scheduled Admission—Addition of a Study Protocol to Test the Existence of Effects of COVID-19 in the Same Study in the Post-COVID-19 Period"

_nutrients, 2024, doi:10.3390/nu16142327_

Round 1

Reviewer 1 Report

Comments and Suggestions for Authors

The manuscript describes an interesting concept. An analysis of meal consumption after pandemi COVID-19 related to mortality may provide some insight. It would still be necessary to state what meals were served to patients at that time. Low meal intake may have been related to unpalatable food. What additional burdens the patients had with other disease entities. Very interesting idea of 4 seasons but seems too small why only one Thursday and not 2 or 3 Thursdays of each season? What is the suggestion to improve eating in patients after COVID-19? Literature sufficient analysis of results good

Author Response

Responses to requests or comments of Reviewer 1

We would like to express our sincere gratitude for the valuable suggestions you have given us regarding our submitted paper. We will provide answers (A: written in green) to each of your questions (Q: written in blue).

Q: An analysis of meal consumption after pandemic COVID-19 related to mortality may provide some insight.

Q1: It would still be necessary to state what meals were served to patients at that time. Low meal intake may have been related to unpalatable food.

⇒A1: The meals provided to hospitalized patients were regular hospital meals, and there were no particular complaints about the taste. For patients with swallowing disorders with an IDSSI score of 6 or less, we provided swallowing-adjusted meals with food texture adjusted on an individual basis. The numbers shown in the Nutritional parameters – “Food texture by IDDSI” section in Tables 1 and 2 are the number of patients who received regular meals, and the remaining patients were provided with swallowing-adjusted meals.

Q2: What additional burdens the patients had with other disease entities.

⇒A2: The primary diagnoses of the patients at the time of admission were classified into WHO disease classification categories I to XXI and compared. Food consumption was set at 75% and 50% cutoff values, and the results of the comparison between the two groups are shown in Supplementary Table 2.

Q3: Very interesting idea of 4 seasons but seems too small why only one Thursday and not 2 or 3 Thursdays of each season?

A3: This idea is not original to us. Originally, the ”nutritionDay" (nDay) survey, which measures the prevalence of malnutrition worldwide, has been conducted once a year on the third Thursday of November, which is proposed by the European and American Clinical Nutrition Societies (ESPEN and ASPEN). Our idea was used as the basis. The study design and references 4-6 are described below.

Q4: What is the suggestion to improve eating in patients after COVID-19? Literature sufficient analysis of results good

A4: Malnutrition due to sarcopenia has also been reported in the post-COVID period [1, 2]. However, it is necessary to verify in the post-COVID period whether this is actually due to the direct effects of COVID, or whether food consumption has actually decreased to less than 75%.

  1. Grund S, Bauer JM. Malnutrition and sarcopenia in COVID-19 survivors. Clin Geriatr Med. 2022; 38(3): 559- 564.
  2. Fyffe I, Sorensen J, Carroll S, MacPhee M, Andrews-Paul A, Crooks VA, et al. Long COVID in long-term care: a rapid realist review. BMJ Open. 2023; 13(12):e076186. doi: 10.1136/bmjopen-2023-076186.

Reviewer 2 Report

Comments and Suggestions for Authors

   The main purpose of the article is to examine the association between in-hospital food consumption and hospital mortality among patients with scheduled admissions. The study aims to determine if consuming less than 75% of the hospital food requirement is linked to higher in-hospital mortality rates. It also introduces a study protocol to assess whether the effects of COVID-19 have influenced in-hospital mortality in the post-pandemic period. The research seeks to provide baseline data from the pre-pandemic period to compare with the post-pandemic period, focusing on the potential long-term impact of COVID-19 on hospitalized patients' mortality related to their food intake.

Here are a few suggestions to improve the paper:

Expand the introduction to provide a more comprehensive background on the importance of hospital food consumption and its relationship to patient outcomes.

Provide more detailed information on the methodology, including how patients were selected, how food consumption was measured, and how in-hospital mortality was defined and recorded.

Address the sample size more explicitly. Discuss whether the sample size was sufficient to detect meaningful differences and how representative the sample is of the broader patient population.

Discuss the practical implications of the findings for clinical practice and hospital nutrition policies.

Include more detailed information on ethical considerations, including how patient consent was obtained and how data privacy was maintained.

Include a more thorough literature review that contextualizes the study within existing research. For example, this paper https://doi.org/10.5603/cj.a2021.0159 investigates the long-term effects of COVID-19, specifically pulmonary arterial hypertension and right ventricular dysfunction in COVID-19 survivors. This is relevant for the article under review, which aims to understand the impacts of COVID-19 on in-hospital mortality.

To reference this study could be relevant to highlight the shared symptoms and challenges faced by COVID-19 survivors, reinforcing the need to study in-hospital mortality and nutritional intake in this population.

By addressing these areas, the paper will be more comprehensive, transparent, and impactful, providing clearer insights and stronger evidence for the relationship between hospital food consumption and patient outcomes.

Author Response

Responses to requests or comments of Reviewer 1

We would like to express our sincere gratitude for the valuable suggestions you have given us regarding our submitted paper. We will provide answers (A: written in green) to each of your questions (Q: written in blue).

Q1 - Expand the introduction to provide a more comprehensive background on the importance of hospital food consumption and its relationship to patient outcomes. A1. According to your kind suggestion, we expanded our introduction to enhance the importance food consumption associated with outcomes, including mortality and the length of hospitalization and added the followings in the “1.1. Findings of the previous study, which showed an association between in-hospital food consumption and in-hospital mortality in patients with NOT scheduled BUT emergency admission” involved in the introduction:

Before completing this study of planned hospital admissions, we paid attention to food consumption in hospitals because of several findings: 1) A 30-day follow-up study of hospitalized patients in 31 units of a Danish university hospital showed that an increased risk of mortality was associated with patients who did not consume 75% of the estimated requirement (energy: OR ¼ 8.08 [1.78; 36.79]; protein: OR ¼ 3.40 [0.74; 15:53]) [2]. 2) Similar findings have been reported in patients with chronic obstructive pulmonary disease (COPD) [3]. 3) This may be due to increased inflammation, which is closely associated with poor appetite and reduced food intake [4].

Consistent with our expectation that hospital food consumption is associated with outcomes,

And added another three references as the follows:

  1. Mikkelsen S, Frost KH, Engelbreth EM, Nilsson L, Peilicke KM, Tobberup R, Holst M. Are nutritional sufficiency of 75% energy and protein requirements relevant targets in patients at nutritional risk? - A one month follow-up study. Clin Nutr ESPEN. 2023; 54: 398-405. doi: 10.1016/j.clnesp.2023.02.007.
  2. Ingadottir AR, Beck AM, Baldwin C, Weekes CE, Geirsdottir1 OG, Ramel A , Gislason T, Gunnardottir I. Association of energy and protein intakes with length of stay, readmission and mortality in hospitalised patients with chronic obstructive pulmonary disease. Br J Nutr. 2018; 119: 543-51. doi: 10.1017/S0007114517003919.
  3. Pourhassan M, Böttger S, Janssen G, Sieske L, Wirth R. The association of inflammation with food intake in older hospitalized patients. J Nutr Health Aging. 2018; 22(5): 589-93. doi: 10.1007/s12603-017-0976-2.

Q2- Provide more detailed information on the methodology, including how patients were selected, how food consumption was measured, and how in-hospital mortality was defined and recorded.

A2. What you kindly pointed out are all important to define the words, we added explanations in the participants session as the follows:

1, how patients were selected: all inpatients staying in hospital on the study day were eligible for the study.

2, how food consumption was measured: Food consumption measurement was conducted by the ward nurses who collected the completed meals, they assessed the amount of food consumption by visually estimating 100, 75, 50 and 25%.

3, how in-hospital mortality was defined and recorded: In-hospital mortality was defined as death at the time of discharge.

Q3- Address the sample size more explicitly. Discuss whether the sample size was sufficient to detect meaningful differences and how representative the sample is of the broader patient population.

A3: From the total number of subjects, 671, subjects who met the exclusion criteria were excluded, and the remaining 583 subjects were used for validation (Figure 1). Generally, the population is 1000, and the required sample size with a margin of error of ±5% is 279, so this is by no means a sufficient number of subjects, and therefore a limitation was written as the follows:

the number is not appropriately enough to draw the conclusive findings. Further large prospective studies on other weekdays or weekends are needed.

Regarding the representativeness of the sample for the broader patient population, the entities are almost as broad as the ICD-10 version, as shown in Supplementary Table 2. This suggests that the entities of the subjects could be representative for additive analyses.

Q4- Discuss the practical implications of the findings for clinical practice and hospital nutrition policies.

A4. According to your excellent suggestion, we added the following proposal in the discussion part:

Based on the results of this study, we propose that in the future, all hospitalized patients who consume less than 75% of the hospital food provided should be considered nutritionally at risk, i.e., patients at high risk for high in-hospital mortality. We propose to develop a nutritional policy that examines 1) the reasons for low food consumption, 2) If food consumption above 75% is not possible, worsening inflammatory symptoms must be recognized and treated earlier to prevent in-hospital mortality.

Q5- Include more detailed information on ethical considerations, including how patient consent was obtained and how data privacy was maintained.

A5. According to your suggestion, we stated the following in the Institutional Review Board Statement:

In order to obtain ethics committee approval, an opt-out procedure was published in the hospital and on the website, stating that patients admitted during the study period who did not wish to participate should inform the hospital of their wishes.

Q6 - Include a more thorough literature review that contextualizes the study within existing research. For example, this paper https://doi.org/10.5603/cj.a2021.0159 investigates the long-term effects of COVID-19, specifically pulmonary arterial hypertension and right ventricular dysfunction in COVID-19 survivors. This is relevant for the article under review, which aims to understand the impacts of COVID-19 on in-hospital mortality. To reference this study could be relevant to highlight the shared symptoms and challenges faced by COVID-19 survivors, reinforcing the need to study in-hospital mortality and nutritional intake in this population. By addressing these areas, the paper will be more comprehensive, transparent, and impactful, providing clearer insights and stronger evidence for the relationship between hospital food consumption and patient outcomes.

A6. According tou your kind suggestion, we added the following sentence in the proposal of study protocol:

As a post-COVID effect, a decline in cardiopulmonary function due to pulmonary hypertension and right ventricular dysfunction [*-***] has been reported. These may well lead to a decrease in food intake during hospitalization. However, no studies have been conducted on food consumption < 75% as an outcome or on in-hospital mortality.

25, Rossi R, Coppi F, Monopoli DE , Sgura FA, Arrotti S, Alfredo F. Pulmonary arterial hypertension and right ventricular systolic dysfunction in COVID-19 survivors. Cardiol J. 2022; 29(1): 163-5. doi: 10.5603/CJ.a2021.0159.

26, Riou M, Coste F, Meyer A, Enache I, Talha S, Charloux A, Reboul C, Geny B. Mechanisms of Pulmonary Vasculopathy in Acute and Long-Term COVID-19: A Review. Int J Mol Sci. 2024 Apr 30;25(9):4941. doi: 10.3390/ijms25094941.

27, Eroume À Egom E, Shiwani HA, Nouthe B. From acute SARS-CoV-2 infection to pulmonary hypertension. Front Physiol. 2022 Dec 19;13:1023758. doi: 10.3389/fphys.2022.1023758.